# Antibiotic treatment failure in children aged 1 to 59 months with World Health Organization-defined severe pneumonia in Malawi: A CPAP IMPACT trial secondary analysis

**Tisungane Mvalo** [1,2]*, **Andrew G. Smith**[3], **Michelle Eckerle**[4,5], **Mina C. Hosseinipour** [1,6], **Davie Kondowe**[1], **Dhananjay Vaidya** [7], **Yisi Liu**[8], **Kelly Corbett**[9], **Dan Nansongole**[1], **Takondwa A. Mtimaukanena**[1], **Norman Lufesi**[10], **Eric D. McCollum**[11,12]

**1** University of North Carolina Project Malawi, Lilongwe, Malawi, **2** Department of Pediatrics, School of Medicine, University of North Carolina at Chapel Hill, Chapel Hill, NC, United States of America, **3** Division of Pediatric Critical Care, Department of Pediatrics, University of Utah, Salt Lake City, Utah, United States of America, **4** Department of Pediatrics, University of Cincinnati College of Medicine, Cincinnati, OH, United States of America, **5** Division of Pediatric Emergency Medicine, Cincinnati Children's Hospital, Cincinnati, OH, United States of America, **6** Division of Infectious Disease, Department of Medicine, School of Medicine, University of North Carolina at Chapel Hill, Chapel Hill, NC, United States of America, **7** Department of Medicine, Epidemiology and the BEAD Core, Johns Hopkins University, Baltimore, MD, United States of America, **8** Department of Pediatrics and the BEAD Core, Johns Hopkins University, Baltimore, MD, United States of America, **9** Department of Pediatrics, Section of Critical Care Medicine, Dartmouth-Hitchcock Medical Center, Lebanon, NH, United States of America, **10** Malawi Ministry of Heath, Lilongwe, Malawi, **11** Eudowood Division of Pediatric Respiratory Sciences, Department of Pediatrics, Global Program for Pediatric Respiratory Sciences, Johns Hopkins School of Medicine, Baltimore, MD, United States of America, **12** Department of International Health, Bloomberg School of Public Health, Johns Hopkins University, Baltimore, MD, United States of America

* tmvalo@unclilongwe.org

## Abstract

### Background

Pneumonia is a leading cause of mortality in children <5 years globally. Early identification of hospitalized children with pneumonia who may fail antibiotics could improve outcomes. We conducted a secondary analysis from the Malawi CPAP IMPACT trial evaluating risk factors for antibiotic failure among children hospitalized with pneumonia.

### Methods

Participants were 1–59 months old with World Health Organization-defined severe pneumonia and hypoxemia, severe malnutrition, and/or HIV exposure/infection. All participants received intravenous antibiotics per standard care. First-line antibiotics were benzylpenicillin and gentamicin for five days. Study staff assessed patients for first-line antibiotic failure daily between days 3–6. When identified, patients failing antibiotics were switched to second-line ceftriaxone. Analyses excluded children receiving ceftriaxone and/or deceased by hospital day two. We compared characteristics between patients with and without treatment failure and fit multivariable logistic regression models to evaluate associations between treatment failure and admission characteristics.

**Data Availability Statement:** All relevant data are within the manuscript and its Supporting Information files.

**Funding:** EDM: The Bill & Melinda Gates Foundation (OPP1123419) (https://www. gatesfoundation.org/) and a CIPHER grant from the International AIDS Society (141022) (https://www. iasociety.org/HIV-Programmes/Programmes/ Paediatrics-CIPHER/CIPHER-Grant-Programme.) The sponsor had no role in study design, data collection and analysis, decision to publish, or preparation of the manuscript.

**Competing interests:** The authors have declared that no competing interests exist.

## Results

From June 2015–March 2018, 644 children were enrolled and 538 analyzed. Antibiotic failure was identified in 251 (46.7%) participants, and 19/251 (7.6%) died. Treatment failure occurred more frequently with severe malnutrition (50.2% (126/251) vs 28.2% (81/287), p<0.001) and amongst those dwelling ≥10km from a health facility (22.3% (56/251) vs 15.3% (44/287), p = 0.026). Severe malnutrition occurred more frequently among children living ≥10km from a health facility than those living <10km (49.0% (49/100) vs 35.7% (275/428), p = 0.014). Children with severe malnutrition (adjusted odds ratio (aOR) 2.2 (95% CI 1.52, 3.24), p<0.001) and pre-hospital antibiotics ((aOR 1.47, 95% CI 1.01, 2.14), p = 0.043) had an elevated aOR for antibiotic treatment failure.

## Conclusion

Severe malnutrition and pre-hospital antibiotic use predicted antibiotic treatment failure in this high-risk severe pneumonia pediatric population in Malawi. Our findings suggest addressing complex sociomedical conditions like severe malnutrition and improving pneumonia etiology diagnostics will be key for better targeting interventions to improve childhood pneumonia outcomes.

## Introduction

Pneumonia is the leading infectious cause of illness and mortality in children under 5 years old globally, with an estimated 900,000 deaths annually [1]. The highest pneumonia burden occurs in low-income and middle-income countries (LMICs), with deaths in Africa representing half of global child pneumonia mortality [1, 2]. This trend continues despite recent reductions in childhood pneumonia cases following the introduction of conjugate vaccines against *Streptococcus pneumoniae* (PCV) and *Haemophilus influenzae* type B (Hib). In LMICs, community acquired pneumonia is clinically diagnosed according to World Health Organization (WHO) Integrated Management of Childhood Illness (IMCI) guidelines [3, 4]. Children with fast breathing and/or chest indrawing but without danger signs (i.e., WHO-defined non-severe pneumonia) are treated with oral antibiotics whilst patients with WHO-defined severe pneumonia are hospitalized for intravenous antibiotics and supportive management [4].

To further reduce pneumonia morbidity and mortality, the early identification of children with pneumonia likely to fail antibiotics and have poor outcomes may be an important care strategy [5]. Early identification can aid health care workers to streamline triage and resource-intensive monitoring and/or treatments to those at greater risk of poor outcomes whilst safely shortening hospital stays for lower risk children. Altogether this could reduce healthcare costs and improve patient outcomes, including mortality [6]. Poor pediatric pneumonia outcomes have been previously associated with young age, hypoxemia, severe acute malnutrition (SAM), HIV infection or exposure, severe disease on hospitalization, low birth weight or prematurity, and chest radiograph consolidation [5–12]. Improved access and reduced travel time to health facilities is also key to successful childhood pneumonia management [13]. While child pneumonia treatment failure definitions vary in the literature [7], generally WHO guidelines recommend switching to a second-line antibiotic regimen in the absence of clinical improvement or with clinical deterioration after 48 hours of first-line antibiotics [3, 6].

Malawi has a high HIV prevalence rate with 1 in every 10 adults living with HIV [14], as well as high levels of poverty and low food security levels with just 8% of 6–23 month olds consuming an adequate diet [15, 16]. Over the past two decades important gains have been made towards addressing some of the previously identified risk factors for treatment failure. HIV incidence rates in Malawi have decreased by at least 60% and perinatal HIV transmission has neared elimination, reducing childhood HIV infection prevalence and exposure [17, 18]. For children infected with HIV, successful cotrimoxazole preventative therapy also reduced severe pneumonia incidence [19]. Furthermore, PCV and Hib vaccines are likely shifting lower respiratory infection etiology towards viruses and/or other non-pneumococcal and haemophilus bacterial causes. For example, in a 2016 cohort of Malawian children with pneumonia and high immunization rates, bacteria were isolated in only 2.6% while viruses in 90.7% of cases [20].

As a result of these notable gains and shifts in socio-demographic characteristics, previously described child pneumonia risk factors may also have evolved. To assess this, we conducted a secondary analysis of data from the Malawi CPAP IMPACT trial to re-evaluate risk factors for antibiotic treatment failure among children hospitalized with WHO-defined severe pneumonia.

## Methods

### Study site

Patients were recruited into this trial from Salima district hospital in Salima, a lakeshore district in the central region of Malawi. This secondary level hospital serves as a referral center for all primary health facilities in Salima district. It has a 250 total bed capacity with approximately 6,500 pediatric medical hospitalizations annually, has no intensive care unit facilities, and serves a population of about 480,000.

### Study population and procedures

This secondary analysis was performed among children aged 1–59 months enrolled into the CPAP IMPACT trial between June 2015 to March 2018. CPAP IMPACT was an individually randomized controlled, open label trial whose primary aim was to investigate the efficacy of bubble continuous positive airway pressure (bCPAP) compared to standard of care low flow oxygen (subsequently referred to as oxygen) on the mortality of children with WHO-defined severe pneumonia. Children with WHO-defined severe pneumonia and one or more co-morbidities including HIV exposure (but HIV-uninfected), HIV infection, or SAM, or the complication of hypoxemia were eligible, as described previously [21, 22]. All trial participants received standardized treatment for severe pneumonia that included intravenous antibiotics. First-line intravenous antibiotics were a 5-day course of benzylpenicillin (50,000 units 6 hourly) and gentamicin (7.5mg/kg once daily), per WHO and national guidelines. Patients enrolled with clinical features suggesting meningoencephalitis (Blantyre coma score <5 and/ or convulsions) were commenced on ceftriaxone 100mg/kg once daily. Patients commenced on ceftriaxone at hospital admission were excluded from this analysis since ceftriaxone was also a second-line antibiotic for severe pneumonia. Additional exclusion criteria included death before day three, withdrawal from CPAP IMPACT trial, or loss-to-follow-up.

Nutritional assessment was performed at admission and involved measuring weight, length (if age < 24 months), height (if age ≥ 24 months), mid-upper arm circumference (MUAC) and assessment for bilateral edema. A classification of SAM required either the presence of a weight for height/length Z-score of < -3, a MUAC of <11.5cm and/or bilateral edema [4]. Moderate acute malnutrition classification was defined by a weight for height/length Z-score

of >-3 to ≤ -2, and/or a MUAC of >11.5cm to ≤ 12.5cm. Blood tests on hospitalization included a malaria rapid diagnostic test (mRDT, SD Bioline), hemoglobin (Hemocue 301+) and HIV rapid test (HIV Determine). Per Malawi guidelines children below 12 months old underwent HIV PCR testing (Abbott m2000). Patients with a hemoglobin < 5g/dL were classified as having severe anemia.

The study team was available 24 hours per day, including weekends, to screen, enroll and clinically review study patients. This team was comprised of non-physician clinicians called clinical officers, and nurses and vital signs assistants. A study pediatrician was available for daily telephonic consults. Study patients were scheduled for reviews by a study clinician twice daily, but were reviewed more often if clinically deteriorating. Patients were assessed for the primary outcome of antibiotic treatment failure during each review from day 3 to day 6 (see Table 1 for primary outcome definition). If treatment failure to the first-line intravenous antibiotic regimen of benzylpenicillin and gentamicin was identified, patients were switched to the second line intravenous antibiotic regimen of ceftriaxone for 5 days at a dose of 80mg/kg once daily. This approach was consistent with recommended WHO and national guidelines for children hospitalized with pneumonia. Antibiotic treatment failure occurring on days 3–5 was defined as 'early treatment failure' whilst treatment failure identified on day 6 was defined as 'late treatment failure.'

In-person supervision visits at the study site were conducted by the study pediatrician every fortnight. The study Principal Investigator and two other key Co-Investigators also performed in person supervision visits at least twice a year. Annual protocol and study procedure refresher trainings were conducted annually and the REDCAP system contained clinical decision trees and data validation prompts promoting quality control. In addition, data entered was reviewed by the study coordinator for quality checks. During the conduct of the study, a clinical research associate performed a monitoring visit at the study site twice.

All infants exposed to HIV or living with HIV were also commenced on high dose cotrimoxazole (21-day course) and prednisolone (5-day course) on admission for presumed *pneumocystis jirovecii* pneumonia. Those aged 12–59 months were commenced on the same high dose cotrimoxazole regimen when identified as having treatment failure from day 3.

**Table 1. Antibiotic treatment failure definitions.**

| |
|---|
| (A) **Early (Day 3–5) treatment failure** |
| *1- Axillary temperature ≥38° Celsius (fever) or* |
| *2- Fever and persistent respiratory danger signs (SpO$_2$ <90%, grunting, head nodding, nasal flaring, severe chest indrawing, very fast breathing (≥70 breaths/minute if 1–11 months; ≥60 breaths/minute if 12–59 months of age), stridor in a calm child, or apnea) or* |
| *3- Fever and persistent general danger signs (inability to eat or drink, lethargy or unconscious, convulsions, vomiting everything) or* |
| *4- New respiratory danger sign or* |
| *5- New general danger sign or* |
| *6- Death during hospitalization and prior to pneumonia cure (resolution of respiratory danger signs and fever), or continued need for oxygen or bCPAP by day 5* |
| (B) **Late (Day 6) treatment failure** |
| *1- Axillary temperature ≥38° Celsius (fever) or* |
| *2- Any respiratory danger sign (new or persistent) or* |
| *3- Any general danger sign (new or persistent) or* |
| *4- Oxygen or bCPAP or* |
| *5- Death on day 6* |

### Data collection

In CPAP IMPACT, study clinicians and nurses prospectively entered data in real time into an electronic REDCAP database on an encrypted tablet when clinically evaluating participants. Variables of interest for this analysis included age, randomization arm, distance from place of residence to nearest health facility, vaccination status, weight, MUAC, length or height, weight, respiratory rate, peripheral arterial oxyhemoglobin saturation ($SpO_2$), respiratory danger signs, general danger signs and laboratory results for HIV, malaria, and hemoglobin.

### Ethics

CPAP IMPACT was reviewed and approved by the Malawi National Health Science Research Committee (Protocol 1325) and the Johns Hopkins Institutional review board (IRB00055734). The trial was registered with Clinicaltrials.gov (NCT02484183). Written informed consent was obtained from the parent or legally accepted guardian of each child prior to enrollment, randomization, and data collection.

### Data analysis

We evaluated patient characteristics comparing those with and without the primary outcome of antibiotic treatment failure (Table 1) and also assessed antibiotic treatment failure by study arm. Categorical variables were tabulated as numbers and percentages, and differences in distributions were tested using chi-square tests. To explore whether distance from the treatment facility ($<10$ vs $\geq 10$ km) may reflect poor health, and thence be associated with antibiotic treatment failure, we examined other clinical characteristics by reported distance to the nearest treatment facility. We further evaluated the association of the binary outcome of antibiotic treatment failure versus study arm and other patient characteristics by fitting logistic regression models to produce adjusted odds ratios. We evaluated a base model that included the following covariates: age in months, sex, randomization group, hemoglobin ($<5g/dL$, $5-10g/dL$ or $>10g/dL$), presence of general danger signs, and distance to the healthcare facility ($<10$ km or $\geq 10$km). Additional variables were added one at a time to the base model in a forward stepwise manner and the adjusted odds ratios were inspected. If the added variable was conceptually similar to any in the base model (e.g., adding hemoglobin as a continuous variable to the model already with hemoglobin as a categorical variable), then the paired variable was dropped to avoid collinearity and allow interpretation. For hypothesis testing, $p <0.05$ was considered significant, and p values between 0.05 and 0.10 were considered as weak evidence of an association.

## Results

In the CPAP IMPACT trial, 1712 children were screened and 644 were enrolled (Fig 1). We excluded an additional 106 participants from this secondary analysis, most commonly for death prior to day three of hospitalization. Our primary outcome of antibiotic treatment failure was identified in 251/538 (46.7%) children; 213/538 (39.6%) had 'early' and 38/538 (7.1%) had 'late' treatment failure. Hospital mortality occurred in 7.6% (19/251) of children with antibiotic treatment failure.

Children with a weight $<5$kg, MUAC $<11.5$cm, weight for age z-score $<-3$, and a residential distance $\geq 10$km to the nearest health facility had higher rates of antibiotic treatment failure, both early and late (Table 2), compared to children not meeting these criteria. We observed weak evidence of a higher frequency of antibiotic treatment failure in the bCPAP

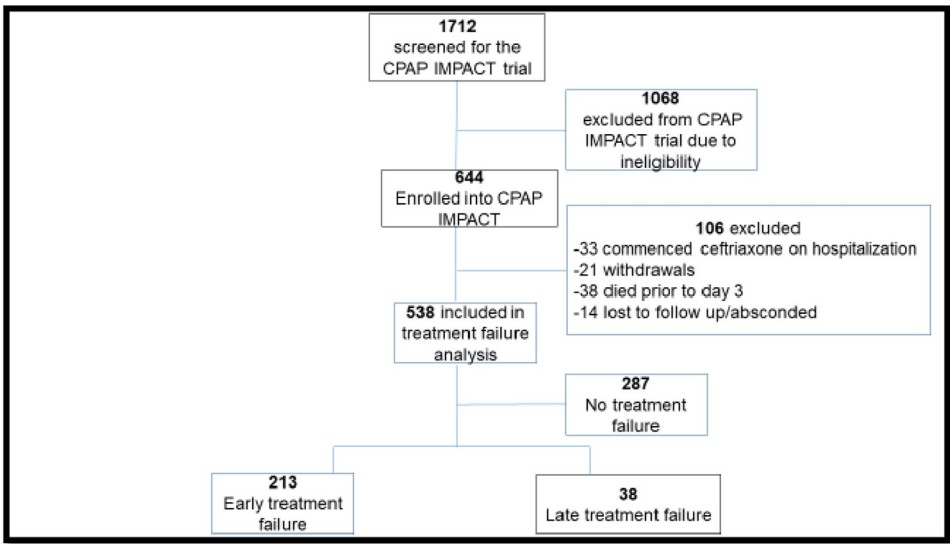

**Fig 1. Consort diagram of CPAP IMPACT study participants analyzed for antibiotic treatment failure.**

arm than the oxygen arm, in HIV infected compared to HIV exposed or uninfected HIV participants, and among children with a hemoglobin <5g/dL compared to ≥5g/dL.

Analysis of the proportion of children on cotrimoxazole preventative therapy according to HIV status revealed that 89% (17/19) of the HIV infected and 69% (69/100) of the HIV uninfected (but exposed) participants were prescribed pre-hospital cotrimoxazole preventative therapy.

Although patients residing ≥10km from the nearest health facility had a higher frequency of treatment failure than children living <10km from a facility, we did not observe any meaningful associations between health facility distance and illness severity at hospital admission (Table 3). However, we did observe a higher proportion of SAM among children residing ≥10km from the nearest facility, compared to <10km (49% (49/100) vs 35.7% (153/428), p = 0.014).

During subgroup analysis of participants having either an *early* or *late* antibiotic treatment failure outcome by study arm, *early* treatment failure cases showed weak evidence for a higher treatment failure frequency in the bCPAP arm than in the oxygen arm [43.9% (111/253) vs 35.8% (102/285), p = 0.056] (Table 4). When examining *early* treatment failure criteria we found that a higher proportion of children receiving bCPAP, compared to oxygen, had fever with persistent respiratory danger signs [35.2% (89/253) vs 27% (77/285), p = 0.041]. We did not find any meaningful associations between *late* treatment failure overall or *late* treatment failure criteria by participant study arm.

We evaluated predictors of antibiotic treatment failure in Table 5. Adjusted analysis revealed the odds of antibiotic treatment failure for children with SAM, compared to without SAM, were 120% higher (aOR 2.2, 95% CI 1.52, 3.24, p<0.001). Use of antibiotics prior to hospitalization, compared to no use, increased the odds of antibiotic treatment failure by 47% (aOR 1.47, 95% CI 1.01, 2.14, p = 0.043). We observed weak evidence of an association between antibiotic treatment failure and participants having at least one general danger sign, participant hemoglobin level at admission, and participant distance to the nearest facility. Notably, neither HIV infection nor HIV exposure without infection were associated with antibiotic treatment failure.

**Table 2. Baseline participant characteristics by antibiotic treatment failure outcome.**

| Characteristic | Variable | No Antibiotic Treatment Failure (N = 287) | Antibiotic Treatment Failure (N = 251) | p-value |
|---|---|---|---|---|
| Study arm | bCPAP | 125 (49.4%) | 128 (50.6%) | 0.084 |
| | oxygen | 162 (56.8%) | 123 (43.2%) | |
| Sex | Females | 129 (52.2%) | 118 (47.8%) | 0.63 |
| SAM | Yes | 81 (39.1%) | 126 (60.9%) | <0.001 |
| | No | 206 (62.2%) | 125 (37.8%) | |
| HIV status | HIV uninfected | 221 (52.7%) | 198 (47.3%) | 0.065 |
| | HIV exposed uninfected | 60 (60%) | 40 (40%) | |
| | HIV infected | 6 (31.6%) | 13 (68.4%) | |
| Malaria* | Positive | 219 (55.3%) | 177 (44.7%) | 0.15 |
| 3 doses of Hib† | No | 12 (54.5%) | 10 (45.5%) | 0.98 |
| | Yes | 119 (56.7%) | 91 (43.3%) | |
| | Unknown | 54 (56.8%) | 41 (43.2%) | |
| 3 doses of PCV†† | No | 11 (61.1%) | 7 (38.9%) | 0.94 |
| | Yes | 118 (56.7%) | 90 (43.3%) | |
| | Unknown | 54 (56.8%) | 41 (43.2%) | |
| Age (months) | 36–59 | 18 (62.1%) | 11 (37.1%) | 0.17 |
| | 12–35 | 81 (58.7%) | 57 (41.3%) | |
| | 1–11 | 188 (50.7%) | 183 (49.3%) | |
| Weight (kg) | ≥15 | 6 (75%) | 2 (25%) | 0.004 |
| | 5-<15 | 225 (57%) | 170 (43%) | |
| | <5 | 56 (41.5%) | 79 (58.5%) | |
| MUAC (cm) | ≥12.5 | 190 (61.3%) | 120 (38.7%) | <0.001 |
| | 11.5-<12.5 | 49 (52.7%) | 44 (47.3%) | |
| | <11.5 | 48 (35.6%) | 87 (64.4%) | |
| Weight for age z-score | ≥-2.0 | 210 (60%) | 140 (40%) | <0.001 |
| | ≥-3 to < -2 | 39 (50%) | 39 (50%) | |
| | <-3.0 | 38 (34.5%) | 72 (65.5%) | |
| Hemoglobin (g/dL)** | >10 | 156 (56.5%) | 120 (43.5%) | 0.069 |
| | 5–10 | 117 (52.5%) | 106 (47.5%) | |
| | <5 | 14 (36.8%) | 24 (63.2%) | |
| Distance to health facility (km) | <10 | 241 (56.3%) | 187 (43.7%) | 0.026 |
| | ≥10 | 44 (44%) | 56 (56%) | |
| | Unknown | 2 (20%) | 8 (80%) | |

Abbreviations: bCPAP: bubble continuous positive airway pressure, SAM: severe acute malnutrition, Hib: *Haemophilus influenzae* type B conjugate vaccine, PCV: Pneumococcal conjugate vaccine, MUAC: mid-upper arm circumference.

* according to malaria rapid diagnostic test, which was missing for 1 child.

** hemoglobin result missing for 1 child.

† Restricted to children aged ≥4 months; 211 children excluded from these as aged <4 months.

††Restricted to children aged ≥4 months; 217 children excluded from these as aged <4 months.

## Discussion

This secondary analysis of the CPAP IMPACT trial revealed higher-risk Malawian children with WHO-defined severe pneumonia and antibiotic treatment failure had high mortality (7.6%), and cases with SAM or pre-hospital antibiotic use had an elevated aOR for antibiotic treatment failure. We also found weaker evidence supporting associations between a higher aOR for antibiotic treatment failure and WHO-defined general danger signs at hospitalization, admission hemoglobin levels, and participant distance from the nearest health facility.

**Table 3. Clinical signs of severe pneumonia and malnutrition at the time of hospitalization according to participant distance from the nearest health facility.**

| Clinical sign | Variable | Distance <10km | Distance ≥10km | p-value |
|---|---|---|---|---|
| | | N = 428 | N = 100 | |
| | | N, (%) | N, (%) | |
| SpO₂ <90% | No | 147 (34.3%) | 41 (41.0%) | 0.21 |
| | Yes | 281 (65.7%) | 59 (59.0%) | |
| SpO₂ 90–95% | No | 356 (83.2%) | 85 (85.0%) | 0.66 |
| | Yes | 72 (16.8%) | 15 (15.0%) | |
| Very fast breathing* | No | 385 (90.0%) | 85 (85.0%) | 0.15 |
| | Yes | 43 (10.0%) | 15 (15.0%) | |
| Presence of at least one respiratory danger sign | No | 204 (47.7%) | 56 (56.0%) | 0.13 |
| | Yes | 224 (52.3%) | 44 (44.0%) | |
| ≥3 respiratory danger signs | No | 391 (91.4%) | 87 (87.0%) | 0.18 |
| | Yes | 37 (8.6%) | 13 (13.0%) | |
| ≥1 general danger sign | No | 383 (89.5%) | 89 (89.0%) | 0.89 |
| | Yes | 45 (10.5%) | 11 (11.0%) | |
| SAM | No | 275 (64.3%) | 51 (51.0%) | 0.014 |
| | Yes | 153 (35.7%) | 49 (49.0%) | |

Abbreviations: SAM, severe acute malnutrition; SpO₂, peripheral arterial oxyhemoglobin saturation.

* Respiratory rate of ≥80/min for children aged less than 2 months, ≥70/min for children aged 2 months to <12 months, and ≥60 for children aged 12 months to <60 months.

The primary analysis of CPAP IMPACT revealed bCPAP, compared to oxygen, did not reduce hospital mortality [22]. The main analysis of treatment failure differed in that it did not distinguish between early versus late antibiotic treatment failure, did not consider the persistence of respiratory or general danger signs alone at day 6, considered fewer variables, and included a wider sample of children including those initiated on ceftriaxone at admission or who died within two days of hospitalization. This focused secondary analysis specifically addresses whether we can identify patients at hospital admission who may benefit from earlier use of second line intravenous antibiotics. Identifying such patients may help reorganized prevention efforts and hospital triage, inpatient monitoring, and use of resource-intensive care.

When putting our results into context it is important to recognize that there is substantial historical variation in published pneumonia treatment failure definitions. Pneumonia treatment failure has been broadly defined as an inadequate response to antimicrobial therapy with some definitions using a 48 hour timepoint whilst others using 72 hours [3, 23]. The treatment failure definitions vary also in relation to either the onset or persistence of varying combinations of clinical criteria usually including clinical discretion for any regimen changes [7]. The WHO guidelines recommend ampicillin/penicillin with gentamicin as first-line intravenous antibiotics for severe pneumonia, and changing to second-line antibiotics without improvement 48 hours after hospitalization [3]. The CPAP IMPACT trial applied a similar approach to WHO guidelines. However, WHO does not specify the exact clinical criteria needed for antibiotic treatment failure. In the programmatic context this ambiguity likely creates variation in the timing of antibiotic changes as well as possible over-use of second-line drugs [24]. Overall, our antibiotic treatment failure rates were high as about half of CPAP IMPACT participants met our study definition, with a case fatality rate amongst those with treatment failure of 7.6%. While high, these rates are comparable to the 50.4% treatment failure and 4.8% case fatality rates observed in a pediatric pneumonia population with similar comorbidities in Tanzania [6].

**Table 4. Criteria for antibiotic treatment failure by study arm.**

| Variable | bCPAP arm | Oxygen arm | p-value |
|---|---|---|---|
| | N = 253 | N = 285 | |
| *Early* (days 3–5) antibiotic treatment failure | | | |
| Yes | 111 (43.9%) | 102 (35.8%) | 0.056 |
| *Early* antibiotic treatment failure criteria | | | |
| fever + requiring bCPAP/ oxygen | | | |
| Yes | 102 (40.3%) | 94 (33.0%) | 0.078 |
| fever + persistent respiratory danger signs | | | |
| Yes | 89 (35.2%) | 77 (27.0%) | 0.041 |
| persistent general danger signs | | | |
| Yes | 29 (11.5%) | 31 (10.9%) | 0.83 |
| persistent respiratory danger signs | | | |
| Yes | 230 (90.9%) | 261 (91.6%) | 0.78 |
| fever + persistent general danger signs | | | |
| Yes | 18 (7.1%) | 20 (7.0%) | 0.97 |
| new respiratory danger signs | | | |
| Yes | 21 (8.3%) | 21 (7.4%) | 0.69 |
| new general danger signs | | | |
| Yes | 7 (2.8%) | 5 (1.8%) | 0.43 |
| death during days 3–5 of hospitalization | | | |
| Yes | 8 (3.2%) | 7 (2.5%) | 0.91 |
| *Late* (day 6) antibiotic treatment failure | | | |
| Yes | 59 (23.3%) | 51 (17.9%) | 0.12 |
| *Late* antibiotic treatment failure criteria | | | |
| Fever (axillary temperature ≥38 degrees Celsius) | | | |
| Yes | 14 (5.5%) | 15 (5.3%) | 0.89 |
| Presence of any respiratory danger sign or general danger sign | | | |
| Yes | 46 (18.2%) | 40 (14.0%) | 0.19 |
| Continued need for oxygen or bCPAP treatment | | | |
| Yes | 55 (21.7%) | 48 (16.8%) | 0.15 |
| Death on day 6 | | | |
| Yes | 2 (0.8%) | 2 (0.7%) | 0.91 |

Abbreviations: bCPAP: bubble continuous positive airway pressure.

Our findings also highlight both the importance and complexity of SAM in children with pneumonia. Children with SAM had 2.5-fold higher aOR for antibiotic treatment failure than those without SAM. Similar trends have been noted in other studies in Africa and Asia [12, 25–27]. Children with SAM are especially challenging to clinically manage in resource-constrained settings, as the condition can cause depressed immunological responses and bacterial gastrointestinal overgrowth, which can increase the risk of gram-negative sepsis. Endotoxins and oxidative stress from gram-negative bacterial sepsis may be a causal pathway leading to elevated antibiotic treatment failure risk in SAM patients [25, 28]. In SAM, the emergence of clinical features of severe pneumonia may also be delayed, resulting in later hospital presentation and worse outcomes [29]. Additionally, SAM also causes multi-organ dysfunction that can reduce the body's ability to cope with insults such as infections, leading to a higher risk of poor outcomes [30].

**Table 5. Predictors of antibiotic treatment failure.**

| Patient variable (N = 538) | | Unadjusted OR (95% CI) | p-value | Adjusted OR (95% CI) | p-value |
|---|---|---|---|---|---|
| **Continuous variables** | | | | | |
| Age (months) | | 0.99 (0.98–1.00) | 0.137 | 0.99 (0.97–1.00) | 0.150 |
| Weight (kg) | | 0.88 (0.83–0.94) | <0.001 | 0.93 (0.81–1.07) | 0.302 |
| Weight for height (z-score) | | 2.08 (1.29–3.36) | 0.003 | 1.02 (0.56–1.83) | 0.957 |
| MUAC (cm) | | 0.77 (0.70–0.85) | <0.001 | 0.82 (0.70–0.96) | 0.012 |
| Hemoglobin (g/dL) | | 0.90 (0.84–0.97) | 0.004 | 0.93 (0.86–1.00) | 0.057 |
| Distance to nearest health facility (km) | | 1.06 (1.01–1.10) | 0.008 | 1.04 (1.00–1.09) | 0.055 |
| **Categorical values** | | | | | |
| **Social-demographic features** | | | | | |
| Distance to nearest health facility (km) | <10 (N = 428) | 1.0 (Reference) | | | |
| | ≥10 (N = 100) | 1.64 (1.06–2.54) | 0.027 | 1.43 (0.90–2.26) | 0.130 |
| Age category (months) | 36–59 (N = 29) | 1.0 (Reference) | | | |
| | 12–35 (N = 134) | 1.10 (0.48–2.52) | 0.813 | 1.13 (0.47–2.70) | 0.779 |
| | 1–11 (N = 349) | 1.55 (0.71–3.39) | 0.267 | 1.65 (0.72–3.79) | 0.236 |
| Males (N = 291) | | 0.92 (0.66–1.29) | 0.632 | 0.97 (0.68–1.39) | 0.874 |
| History prematurity (N = 24) | | 1.63 (0.71–3.74) | 0.247 | 0.52 (0.46–2.79) | 0.787 |
| Indoor cooking at home with smoke exposure (N = 103) | | 0.96 (0.62–1.47) | 0.834 | 1.26 (0.80–1.97) | 0.325 |
| Cigarette smoke exposure at home (N = 108) | | 0.71 (0.46–1.09) | 0.117 | 0.84 (0.53–1.31) | 0.435 |
| **Medical history** | | | | | |
| Antibiotics within 7 days prior to hospitalization (N = 193) | | 1.38 (0.97–1.97) | 0.072 | 1.47 (1.01–2.14) | 0.043 |
| History of prior hospitalized pneumonia (N = 135) | | 0.93 (0.63–1.37) | 0.712 | 1.07 (0.70–1.65) | 0.744 |
| Known TB exposure (N = 18) | | 0.56 (0.21–1.50) | 0.247 | 0.61 (0.22–1.73) | 0.353 |
| 3 doses Hib vaccine amongst children aged ≥ 4 months (N = 210) | | 0.92 (0.38–2.22) | 0.849 | 1.32 (0.50–3.46) | 0.575 |
| 3 doses PCV vaccine amongst children aged ≥ 4 months (N = 208) | | 1.20 (0.45–3.21) | 0.719 | 1.70 (0.58–5.02) | 0.334 |
| **Clinical signs and symptoms at the time of hospitalization** | | | | | |
| Severe hypoxemia* (N = 343) | | 0.80 (0.56–1.13) | 0.207 | 0.83 (0.55–1.24) | 0.349 |
| Mild hypoxemia† (N = 90) | | 1.31 (0.83–2.06) | 0.247 | 0.96 (0.59–1.58) | 0.873 |
| Wheeze (N = 117) | | 0.85 (0.57–1.29) | 0.453 | 1.20 (0.76–1.89) | 0.442 |
| Very fast breathing ∫ (N = 184) | | 0.77 (0.54–1.10) | 0.153 | 0.84 (0.58–1.23) | 0.369 |
| ≥1 respiratory danger sign (N = 274) | | 0.77 (0.55–1.08) | 0.127 | 1.70 (0.44–6.52) | 0.442 |
| ≥3 respiratory danger signs (N = 52) | | 0.82 (0.46–1.47) | 0.509 | 0.84 (0.44–1.61) | 0.600 |
| ≥1 general danger sign (N = 56) | | 2.05 (1.16–3.63) | 0.013 | 1.83 (1.0–3.35) | 0.051 |
| **Anthropometrics and nutritional assessment** | | | | | |
| Weight category (kg) | ≥15 (N = 8) | 1.0 (Reference) | | | |
| | 5 to <15 (N = 395) | 2.27 (0.45–11.37) | 0.320 | 1.49 (0.24–9.07) | 0.667 |
| | <5 (N = 135) | 4.23 (0.82–21.74) | 0.084 | 1.51 (0.21–10.68) | 0.681 |
| SAM | Absent (N = 331) | 1.0 (Reference) | | | |
| | Present (N = 207) | 2.56 (1.79–3.66) | <0.001 | 2.22 (1.52–3.24) | <0.001 |
| MUAC (cm) | ≥12.5 (N = 310) | 1.0 (Reference) | | | |
| | 11.5 to <13.5 (N = 93) | 1.42 (0.89–2.27) | 0.140 | 1.27 (0.74–2.16) | 0.384 |
| | <11.5 (N = 135) | 2.87 (1.89–4.37) | <0.001 | 1.83 (0.89–3.78) | 0.103 |
| **Blood tests** | | | | | |
| MRDT | Negative (N = 396) | 1.0 (Reference) | | | |
| | Positive (N = 141) | 1.33 (0.90–1.95) | 0.149 | 1.21 (0.76–1.92) | 0.422 |

(*Continued*)

**Table 5.** (Continued)

| Patient variable | | Unadjusted OR | p-value | Adjusted OR | p-value |
|---|---|---|---|---|---|
| (N = 538) | | (95% CI) | | (95% CI) | |
| Hemoglobin (g/dL) | >10 (N = 276) | 1.0 (Reference) | | | |
| | 5–10 (N = 223) | 1.18 (0.83–1.68) | 0.366 | 1.07 (0.74–1.56) | 0.717 |
| | <5 (N = 38) | 2.23 (1.11–4.49) | 0.025 | 1.46 (0.67–3.18) | 0.340 |
| **HIV status** | | | | | |
| HIV negative unexposed (N = 419) | | 1.0 (Reference) | | | |
| HIV exposed uninfected (N = 100) | | 0.74 (0.48–1.16) | 0.192 | 0.81 (0.51–1.30) | 0.382 |
| HIV infected (N = 19) | | 2.42 (0.92–6.50) | 0.079 | 2.50 (0.83–7.52) | 0.104 |

Abbreviations: bCPAP, bubble continuous positive airway pressure; MUAC, mid-upper arm circumference; MRDT, malaria rapid diagnostic test; SAM, severe acute malnutrition; TB, tuberculosis.

* Peripheral pulse oximetry oxygen saturation <90%.

† Peripheral pulse oximetry oxygen saturation 90%-95%.

∫ Respiratory rate of ≥80/min for children aged less than 2 months, ≥70/min for children aged 2 months but <12 months and ≥60 for children aged 12 months but <60 months.

In our analysis, increased rates of treatment failure were significantly observed in children residing ≥10km from the nearest health facility, although adjusted models found weak evidence of this association. Families with poor socio-economic circumstances and long distances to a hospital may not present to care at all or may present late with advanced disease, as has been reported in studies from Kenya, Namibia and The Philippines [13, 31, 32]. Interestingly, in our analysis we did not find that children residing ≥ 10km from the nearest health facility had more severe pneumonia disease on presentation. In our cohort we instead found increased levels of SAM in children residing further from the nearest health facility. Altogether, this suggests improving the clinical course and outcome of childhood pneumonia in Malawi will require more directly addressing the poor socio-economic factors that place the patient at risk of poor pneumonia outcomes.

Prior use of antibiotics in our cohort was predictive of antibiotic treatment failure. A similar observation was also observed in India [33]. It may be that patients taking oral antibiotics but still requiring hospitalization have pneumonia due to a bacterial pathogen not normally responsive to the first-line drug or the pathogen as acquired new antimicrobial resistance. Poor adherence to oral antibiotic courses for pneumonia is also a challenge and up to 20% of nonadherent pneumonia cases have been reported in Malawi [34]. A poor antibiotic response may also be due to a non-bacterial illness altogether. Indeed, one study on childhood pneumonia etiology in the northern region of Malawi reported the predominance of viral pathogen isolates [20].

Advancing both our understanding and clinical management of treatment failure would be improved with more accurate pneumonia diagnostics. WHO-defined pneumonia is a syndromic condition and it is important to note that a high proportion of patients meeting criteria may not have bacterial pneumonia at all. Children with acute gastroenteritis or septicemia may present with respiratory compensation mimicking clinical pneumonia signs due to an underlying primary metabolic acidosis. A primary metabolic acidosis may present as breathing difficulties making it challenging to differentiate from pneumonia defined only by clinical signs [35]. In this study we found one or more WHO-defined general danger signs were weakly predictive of an increased aOR for antibiotic treatment failure. No other clinical signs were predictive. Studies in Kenya have similarly reported the presence of general danger signs

as associated with an increased risk of pneumonia treatment failure [7]. Our findings overall stress the need for further research on childhood pneumonia etiology, antimicrobial resistance profiles, and improved childhood pneumonia diagnostics.

While increased rates of treatment failure were observed in children with HIV infection, HIV infection and HIV exposure without infection did not predict treatment failure in our cohort. This is likely due to advances in HIV prevention, antiretroviral access, and *Pneumocystis jiroveci* preventative treatment [19, 36]. Also, in contrast with other studies we did not find hypoxemia as associated with treatment failure. This may be due to trial participants having improved access and earlier correction of hypoxemia as compared to other previous studies.

Our analysis had limitations. First, outside of malaria testing, the trial did not investigate pneumonia etiology and it is established that drug resistant pathogens increase treatment failure risk [23]. In LMICs, microbiological investigations for pneumonia are not routinely performed and CPAP IMPACT mirrored this approach. Second, the trial did not record pre-hospital antibiotic regimens limiting more granular analyses. Third, chest radiographs were not performed on enrollment or amongst those with antibiotic treatment failure. At Salima District Hospital, mobile chest radiography was not present and only stable patients could be taken to the radiology department for imaging. This scenario is typical of LMICs. Fourth, CPAP IMPACT lacked cases with less severe disease and this higher disease severity likely contributes to the high treatment failure rate observed.

## Conclusion

In sum, our analysis showed children with SAM had increased rates and elevated risk of antibiotic treatment failure. Causes of SAM are complex and multifactorial, encompassing contextual, societal and economic domains [37]. Policy for reducing childhood pneumonia will need to also address issues like SAM in order to improve pneumonia outcomes. Increased antibiotic treatment failure risk was also noted in children with severe pneumonia who had received prior antibiotics, indicating the need to further investigate childhood pneumonia etiology and susceptibility of pathogens to antibiotics. Lastly, improved diagnostics are needed to refine our understanding of and approaches to antibiotic treatment failure in children with pneumonia in LMICs.

## Supporting information

**S1 File. Data.**
(XLSX)

**S2 File. Data dictionary.**
(CSV)

**S3 File. Codebook.**
(XLSX)

**S4 File. Analysis.**
(DO)

## Acknowledgments

We would like to thank all the children and their caregivers who participated in the CPAP IMPACT trial, Salima District Hospital staff, the Malawi Ministry of Health for their support and the CPAP IMPACT study team.

## Author Contributions

**Conceptualization:** Eric D. McCollum.

**Data curation:** Davie Kondowe, Dan Nansongole, Takondwa A. Mtimaukanena.

**Formal analysis:** Dhananjay Vaidya, Yisi Liu.

**Funding acquisition:** Andrew G. Smith, Eric D. McCollum.

**Project administration:** Tisungane Mvalo, Eric D. McCollum.

**Supervision:** Tisungane Mvalo, Andrew G. Smith, Michelle Eckerle, Eric D. McCollum.

**Writing – original draft:** Tisungane Mvalo.

**Writing – review & editing:** Andrew G. Smith, Michelle Eckerle, Mina C. Hosseinipour, Kelly Corbett, Norman Lufesi, Eric D. McCollum.

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
