## [Decision Letter · Decision Letter 0]

3 Aug 2022

PONE-D-22-10313Antibiotic treatment failure in children aged 1 to 59 months with World Health Organization-defined severe pneumonia in Malawi: a CPAP IMPACT trial secondary analysisPLOS ONE

Dear Dr. Mvalo,

Thank you for submitting your manuscript to PLOS ONE. After careful consideration, we feel that it has merit but does not fully meet PLOS ONE’s publication criteria as it currently stands. Therefore, we invite you to submit a revised version of the manuscript that addresses the points raised during the review process.

The manuscript has been evaluated by three reviewers, and their comments are available below.

The reviewers have raised a number of major concerns. They request improvements to the reporting of methodological aspects of the study such as clarification about the study area. The reviewers also note concerns about the statistical analyses presented and request re-analyses be completed.

Could you please carefully revise the manuscript to address all comments raised?

We look forward to receiving your revised manuscript.

Kind regards,

Lorena Verduci

Staff Editor

PLOS ONE

Journal Requirements:

The Bill & Melinda Gates Foundation (OPP1123419), International AIDS Society (141022) and Health Empowering Humanity provided funding for CPAP IMPACT. 

However, funding information should not appear in the Acknowledgments section or other areas of your manuscript. We will only publish funding information present in the Funding Statement section of the online submission form. 

EDM: The Bill & Melinda Gates Foundation (OPP1123419) (https://www.gatesfoundation.org/) and a CIPHER grant from the International AIDS Society (141022) (https://www.iasociety.org/HIV-Programmes/Programmes/Paediatrics-CIPHER/CIPHER-Grant-Programme.)

The sponsor had no role in study design, data collection and  analysis, decision to publish, or preparation of the manuscript.

5. Please include a separate caption for figure 1 in your manuscript.

Reviewers' comments:

Reviewer's Responses to Questions

**Comments to the Author**

1. Is the manuscript technically sound, and do the data support the conclusions?

Reviewer #1: Yes

Reviewer #2: Yes

Reviewer #3: Yes

2. Has the statistical analysis been performed appropriately and rigorously? 

Reviewer #1: Yes

Reviewer #2: Yes

Reviewer #3: No

3. Have the authors made all data underlying the findings in their manuscript fully available?

Reviewer #1: Yes

Reviewer #2: Yes

Reviewer #3: Yes

4. Is the manuscript presented in an intelligible fashion and written in standard English?

Reviewer #1: Yes

Reviewer #2: Yes

Reviewer #3: Yes

5. Review Comments to the Author

Reviewer #1: The authors conducted a secondary analysis to investigate risk factors for antibiotic failure among children hospitalized with pneumonia. They analyzed 538 children and identified higher odds of treatment failure in children with SAM, pre-hospital antibiotic use, and with general danger sign of hospitalization.

1. Abstract. “Children in this analysis were alive and not receiving ceftriaxone by hospital day 2.” Why exclude these children? Will it be more appropriate to include them as treatment failure?

2. Line 217. “the adjusted adds ratios” has typo?

3. Line 224 “p values between 0.05 and 0.10 indicated a weak trend to association”. Please avoid to use “trend to association” as there might cause confusion about trend test.

4. Table 1. The proportion reported within the study arm may not be appropriate. It would be more appropriate to report the proportion of no treatment failure within bCPAP and so on rather reporting the proportion of bCOAP within no treatment failure group.

5. Table 1. The concern raised in point 4 applies to many other variables. Besides, if follow the logic of the presentation, how come the proportions don’t add up to 1 for some characteristics, 3 doses of hib and so on.

6. Line 273. “each increase of hemoglobin ….decreased the odds ratio…” Hemoglobin is not an intervention so it’s questionable to imply casual effect here. Similar concern applies to other predictors and other texts as well.

Reviewer #2: This paper is very important and is worthy publishing for wider audience especially due to the etiological and epidemiological shifts of Pneumonia cases characteristics due to many interventions done so far in Malawi and beyond.

I have a very few comments which the authors may consider them;

1. Sentence on line number 104-105 is not clear. Please check it

2. I think this paper can best benefit from the recent published on Malawian data (https://www.ncbi.nlm.nih.gov/pmc/articles/PMC8323352/) for instance on introduction paragraph 3 there is no clear supporting evidence of the arguments but this paper can help.

3. In your finding, there is and indication that prior antibiotic intake is a predictor of antibiotic treatment failure, did you capture the details of the type of antibiotic taken?? If yes please provide the details if not the need to include this on the limitation as there is no scientific link or explanation on this finding

4. I feel your discussion is unnecessarily long due to inclusion of some aspects which are not significant finding in your study. You may consider focusing on the key and significant findings

Reviewer #3: Comments

Thank you for invitation to review Antibiotic treatment failure in children aged 1 to 59 months with World Health Organization-defined severe pneumonia in Malawi: a CPAP IMPACT trial secondary analysis. The article is presented in a structured way and is well written. The research has ethical approval and meets the applicable standards for ethics and research integrity. Generally, the paper is well written and structured however in my opinion the paper has shortcomings in regards the abstract, introduction, and methods parts. Below I provide some remarks on the introduction and method parts. Additionally, I suggest the authors to discuss the implication of the result.

1. In the abstract section, avoid use of abbreviations (CPAP, SAM, and OR). Similarly in abstract section, methods are not explanatory, try to indicate method of data analysis.

2. In the abstract section, your conclusion should be based on your findings.

3. In the introduction section, you need to show the previous works, how your finding adds to the existing knowledge, and gap of your work. Furthermore, you need to show the impact of the problem using figures rather than words. Generally, your introduction section lacks figures.

4. On the method part, it is too shallow and it didn’t address all important components of research methodology. You didn’t mention the study period. You need to explain in detail about the study area. How you control possible confounders? You didn’t mention quality control measures. You need to show how you measure your outcome variable in detail.

5. On the result part, I recommend the author to write the outcome variable in detail rather than discussing supporting evidences. I suggest the authors to limit the usage of abbreviations in result section.

6. The discussion is well written and structured. However, it lacks the implication of the results.

6. PLOS authors have the option to publish the peer review history of their article (what does this mean?). If published, this will include your full peer review and any attached files.

Reviewer #1: No

Reviewer #2: **Yes: **Master R.O. Chisale

Reviewer #3: No

---

## [Author Response · Author response to Decision Letter 0]

6 Oct 2022

We would like to thank you for reviewing our manuscript Ref: [PONE-D-22-10313] - [EMID:cf85d3bc2844e620].

 With reference to feedback provided on 3 August 2022, we have made edits to the manuscript in response to the comments to conform the manuscript to the PLOS ONE journal. We have also made edits in relation to comments made by the 3 reviewers.

Please find our responses in the attached "Response to Reviewers" document we have attached to this submission

---

## [Decision Letter · Decision Letter 1]

24 Nov 2022

Antibiotic treatment failure in children aged 1 to 59 months with World Health Organization-defined severe pneumonia in Malawi: a CPAP IMPACT trial secondary analysis

PONE-D-22-10313R1

Dear Dr. Mvalo,

We’re pleased to inform you that your manuscript has been judged scientifically suitable for publication and will be formally accepted for publication once it meets all outstanding technical requirements.

Kind regards,

Maria Elisabeth Johanna Zalm, Ph.D

Editorial Office

PLOS ONE

Additional Editor Comments (optional):

Reviewers' comments:

Reviewer's Responses to Questions

**Comments to the Author**

1. If the authors have adequately addressed your comments raised in a previous round of review and you feel that this manuscript is now acceptable for publication, you may indicate that here to bypass the “Comments to the Author” section, enter your conflict of interest statement in the “Confidential to Editor” section, and submit your "Accept" recommendation.

Reviewer #1: All comments have been addressed

Reviewer #2: All comments have been addressed

Reviewer #3: (No Response)

2. Is the manuscript technically sound, and do the data support the conclusions?

Reviewer #1: (No Response)

Reviewer #2: Yes

Reviewer #3: Yes

3. Has the statistical analysis been performed appropriately and rigorously? 

Reviewer #1: (No Response)

Reviewer #2: Yes

Reviewer #3: Yes

4. Have the authors made all data underlying the findings in their manuscript fully available?

Reviewer #1: (No Response)

Reviewer #2: Yes

Reviewer #3: Yes

5. Is the manuscript presented in an intelligible fashion and written in standard English?

Reviewer #1: (No Response)

Reviewer #2: Yes

Reviewer #3: Yes

6. Review Comments to the Author

Reviewer #1: (No Response)

Reviewer #2: (No Response)

Reviewer #3: Thank you for your effort in addressing the previous comments. Dear authors, you can increase the quality of your paper by revising the typo errors. Try to avoid use of abbreviations in the abstract section and write the full word instead of using its abbreviation.

7. PLOS authors have the option to publish the peer review history of their article (what does this mean?). If published, this will include your full peer review and any attached files.

Reviewer #1: No

Reviewer #2: **Yes: **Master R.O. Chisale

Reviewer #3: No

---

## [Editor Report · Acceptance letter]

5 Dec 2022

PONE-D-22-10313R1 

Antibiotic treatment failure in children aged 1 to 59 months with World Health Organization-defined severe pneumonia in Malawi: a CPAP IMPACT trial secondary analysis 

Dear Dr. Mvalo:

I'm pleased to inform you that your manuscript has been deemed suitable for publication in PLOS ONE. Congratulations! Your manuscript is now with our production department. 

Kind regards, 

on behalf of

Dr. Maria Elisabeth Johanna Zalm 

Staff Editor

PLOS ONE